# Zircon Hf-Isotopic Mapping Applied to the Metal Exploration of the Sanjiang Tethyan Orogenic Belt, Southwestern China

**Bin Du [1,2], Zian Yang [1,*], Lifei Yang [3], Qi Chen [2], Jiaxuan Zhu [2], Kangxing Shi [2], Gao Li [1], Lei Wang [1] and Jia Lu [1]**

[1]  China Non-Ferrous Metals Resource Geological Survey, No.5 Courtyard Area 4, Anwai Beiyuan, Chaoyang District, Beijing 100012, China; dubin6866@126.com (B.D.); bj_ligao@126.com (G.L.); wla2009@sina.cn (L.W.); chnlujia@163.com (J.L.)

[2]  State Key Laboratory of Geological Processes and Mineral Resources, China University of Geosciences, No. 29 Xueyuan Road, Haidian District, Beijing 100083, China; dirk@cugb.edu.cn (Q.C.); 3001190005@cugb.edu.cn (J.Z.); shikx@email.cugb.edu.cn (K.S.)

[3]  School of Earth of Sciences, East China University of Technology, 418 Guanglan Road, Nanchang 330013, China; lfyang@ecut.edu.cn

*   Correspondence: yzaldy88@126.com; Tel./Fax: +86-10-84925807

**Abstract:** Zircon Hf-isotopic mapping can be regarded as a useful tool for evaluating the coupling relationship between lithospheric structure and metallic mineralization. Hence, this method shows important significance for mineral prediction. To explore this potential, the published granite zircon Hf isotope data from the Sanjiang Tethyan Orogen were systematically compiled. This study uses the Kriging weighted interpolation in the Mapgis software system to contour Hf isotopes, revealing a relation between the crustal structure and metallogenesis. The mapping results suggest that the Changning–Menglian suture zone is the boundary between ancient and juvenile crust, viz., the western terranes have ancient crust attributes, whereas the eastern terranes exhibit the properties of new juvenile crust. In addition, this study also found that the mineralization and element types in the Sanjiang Tethyan Orogen have a coupling relationship with the crustal structure. The distribution of porphyry Cu-Mo-Au deposits is mainly controlled by the new juvenile crust, whereas the magmatic-hydrothermal Sn-W and porphyry Mo-W(-Cu) deposits are closely related to the reworked ancient crust. The results of zircon Hf isotope mapping prove that the formation and spatial distribution of deposits are related to the composition and properties of the crust. Hf isotope mapping can reveal the regional metallogenic rules and explore metallogenic prediction and metallogenic potential evaluation.

**Keywords:** Hf isotope mapping; mineral exploration; Sanjiang Tethyan orogenic belt

## 1. Introduction

Standard methods of geological prospecting include regional geological mapping, geophysical [1,2] and geochemical surveys [3,4], remote sensing [5–9], and aerial surveys [10–12]. These traditional geological methods have found many deposits, but they also have serious shortcomings, which are costly and challenging for finding mineral deposits. Therefore, new methods need to be developed to explore potential mineral deposits [13–15].

In recent years, regional isotope tracer mapping methods have been widely used to evaluate accretionary and collisional orogeny processes [16–18], describe regional lithospheric three-dimensional architectures [16,19], constrain geotectonic boundaries [17,18,20–24], define the compositions and properties of deep geotectonic units [25,26], surmise the crustal growth [26], and reveal the distribution of mineral deposits [16–18,20–24,27]. Hence, this method shows important significance in mineral exploration and prediction. The Sanjiang Tethyan orogenic belt is located on the southeastern margin of the Tibetan Plateau, belonging to the combined zone of Gondwana and ancient

Eurasia (Figure 1). It has experienced Proto-, Paleo-, Meso-, and Neo-Tethyan evolution and the subsequent oblique continental collision [17,28–32]. Correspondingly, this belt has formed episodic and diverse metallogeny with the tectonics evolved from Tethyan accretionary orogenesis to collisional orogenesis [32–34]. Therefore, the Sanjiang Tethys domain is a natural laboratory for exploration and prospecting for deposits. However, the evolution and structure of the continental lithosphere that controls the localization of ore deposits still remain poorly understood.

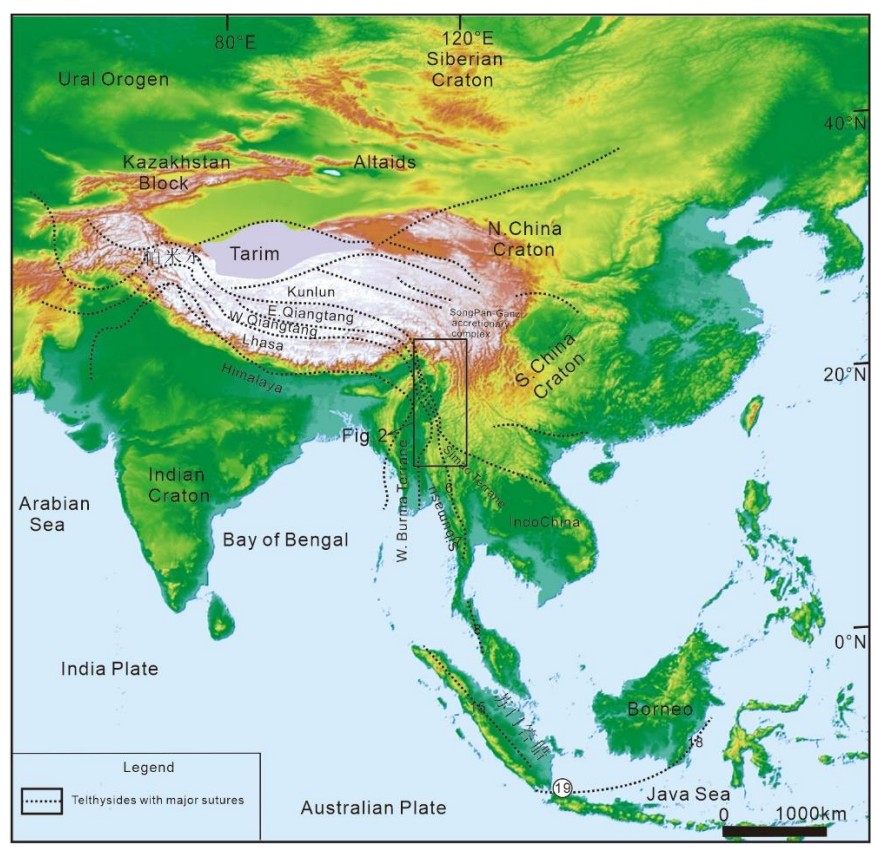

**Figure 1.** Geological map showing the tectonic framework of the Sanjiang Tethyan Orogen and its adjacent areas. Adapted with permission from Ref. [17]. Copyright 2016 ELSEVIER.

In this paper, based on the reviews of the previous zircon Hf-isotopic research, we use the zircon Hf isotope mapping method to explore the genetic relationship between the tectonic evolution and various deposits in Sanjiang Tethyan orogenic belt. This study also summarizes the distribution of different deposits in the area and provides a theoretical basis for the following mineralization and exploration.

## 2. Hf Isotope Mapping's Application to Mineral Prospection

In recent years, some researchers have made more attempts to use zircon Hf isotope mapping to explore the resolution of the three-dimensional tectonic framework and mineralization pattern, mainly focusing on the regions of the Yilgarn Craton of Western Australia, the Lhasa terrane in China, the Eastern Qinling Orogen, and the North China Craton [16,20,22–24,27,35].

Mole et al. [20] performed regional zircon Hf isotopic mapping of the Yilgarn Craton of Western Australia and found that the distribution of magmatic-related deposits is closely related to the formation of material from earth's deep mantle and the new crust. Hou et al. [16] conducted zircon Hf-isotopic mapping of regional magmatic rocks in the Lhasa terrane, revealing that the upper lithosphere material structure and composition of the Lhasa terrane are consistent with the deposit distribution of the Gangdese metallogenic

belt. The porphyry Cu(-Mo-Au) deposits are all distributed in the positive zircon $\varepsilon$Hf(t) area, which is associated with the juvenile crust formation in the South and North Lhasa sub-terrane. Granite-related Pb-Zn deposits are typically localized along the margin of the old crustal block bounded by lithospheric faults. The oldest crustal region developed along the margin of the old crustal block is bounded by lithospheric faults, which is in the negative zone of zircon $\varepsilon$Hf (t). The Hf isotope mappings show that the juvenile crust has a primary control on the formation of porphyry Cu(-Mo-Au) deposits. The distribution of Pb-Zn-Mo deposits in the central sub-terrane is constrained by the inhomogeneity of the composition of ancient crustal blocks and remelting or reworking. Wang et al. [22] revealed that the formation of porphyry and porphyry-skarn Mo(-W) deposits are closely related to the remelting of the ancient crust through zircon Hf isotope mapping of the east Qinling orogen. Wang et al. [23] studied the zircon Hf isotope mapping of the North China Craton and found that the formation of BIF-type iron deposits is associated with the remelting modification of ancient crust, and the formation of orogenic gold deposits is related to the formation of juvenile crust [36]. Deng et al. [24] demonstrated that isotope mapping can constrain the distribution pattern of large-scale gold deposits by conducting zircon Hf isotope mappings of magmatic rocks in the southeast North China Craton. Jiao Dong-type gold deposits and porphyry-skarn Mo(-W-Cu) deposits are typically localized at regions with negative $\varepsilon$Hf(t) values, showing an ancient crustal base composition. Porphyry-skarn Cu(-Au-Mo) deposits in the west of Shandong province cluster variable $\varepsilon$Hf (t) values, showing an ancient crustal and juvenile crustal composition, indicating the contribution of mantle-derived material to mineralization.

## 3. Geologic Setting

The Sanjiang (Three Rivers) region is named due to it being drained by three major rivers: the Jinshajiang, Lancangjiang, and Nujiang [17,30]. The region is the eastern segment of the Tethys-Himalayan tectonic domain, covering the southeastern part of the Tibetan Plateau and the western part of Yunnan Province (Figure 1) [28,30–33,37]. The Sanjiang region lies adjacent to the South China block and Songpan-Garzê accretionary complex in the east and the West Burma block in the west [17,30,31]. It is composed of seven blocks, including the Simao block, Baoshan block, Tengchong block, Zhongza block, East Qiangtang block, West Qiangtang block, and Lhasa block (Figure 2). Some sutures are preserved in the Sanjiang region as evidence of the tectonic evolution of the Tethys Ocean, such as Longmucuo–Shuanghu Changning–Menglian, Jinshajiang, Ailaoshan, Garzê–Litang, and Nujiang (Figure 2). Among them, the Longmucuo–Shuanghu suture, Changning–Menglian suture, Jinshajiang suture, and Ailaoshan suture are all north–south sutures (Figure 2) [18,28,30,38–42].

Along with the process of Tethys accretionary orogeny and collisional orogeny, a series of minerals associated with the Tethys orogenic belt developed in the southwestern Sanjiang Tethys orogenic belt. The main mineral types can be divided into eight categories, including volcanogenic massive sulfide (VMS) Pb-Zn-Cu-Ag deposits, porphyry Cu-(Mo-Au) deposits, porphyry Mo-W(-Cu) deposits, magmatic-hydrothermal Sn-W deposits, magmatic-hydrothermal Pb-Zn-Cu-Ag deposits, Mississippi Valley Type (MVT) Pb-Zn deposits, hydrothermal Au deposits and orogenic Au deposits (Figure 1) [17,34]. The porphyry Cu-(Mo-Au) deposits concentrate in the Jinshajiang–Honghe alkali-rich porphyry Cu-(Mo-Au) deposits belt and the Garzê porphyry Cu-(Mo-Au) deposits belt on the southern edge of the Yidun Island arc [43–45]. The mineralization era of the Jinshajiang–Honghe alkali-rich porphyry Cu-(Mo-Au) deposits belt was concentrated in the Cenozoic and can be divided into the Jinshajiang porphyry belt in the north and the Honghe porphyry belt in the south. The porphyry deposits in the North belt are mainly Cu-(Mo-Au) deposits [46–48]. The porphyry deposits in the south belt are mainly Au-(Cu-Mo) deposits [49–52]. The mineralization age of the Garzê porphyry Cu-(Mo-Au) deposits belt on the southern edge of the Yidun Island Arc is Late Triassic [45,53]. Porphyry Mo-W(-Cu) deposits are developed in the southern margin of the Yidun Island Arc of the Sanjiang orogenic belt and the

contact position is between the Garzê–Litang combined zone and the Xiangcheng–Lijiang land margin depression at the western margin of the South China Craton [45,54]. These Late Cretaceous deposits distributed in a nearly north–south direction are large porphyry Mo-W(-Cu) deposits [45,54–56]. The magmatic-hydrothermal Sn-W deposits are located in the Tengchong–Baoshan block, the northern part of the Yidun volcanic arc, and the Changning–Menglian suture. The age span of the deposits ranges from Early Cretaceous, Late Cretaceous, and Paleocene [29,57].

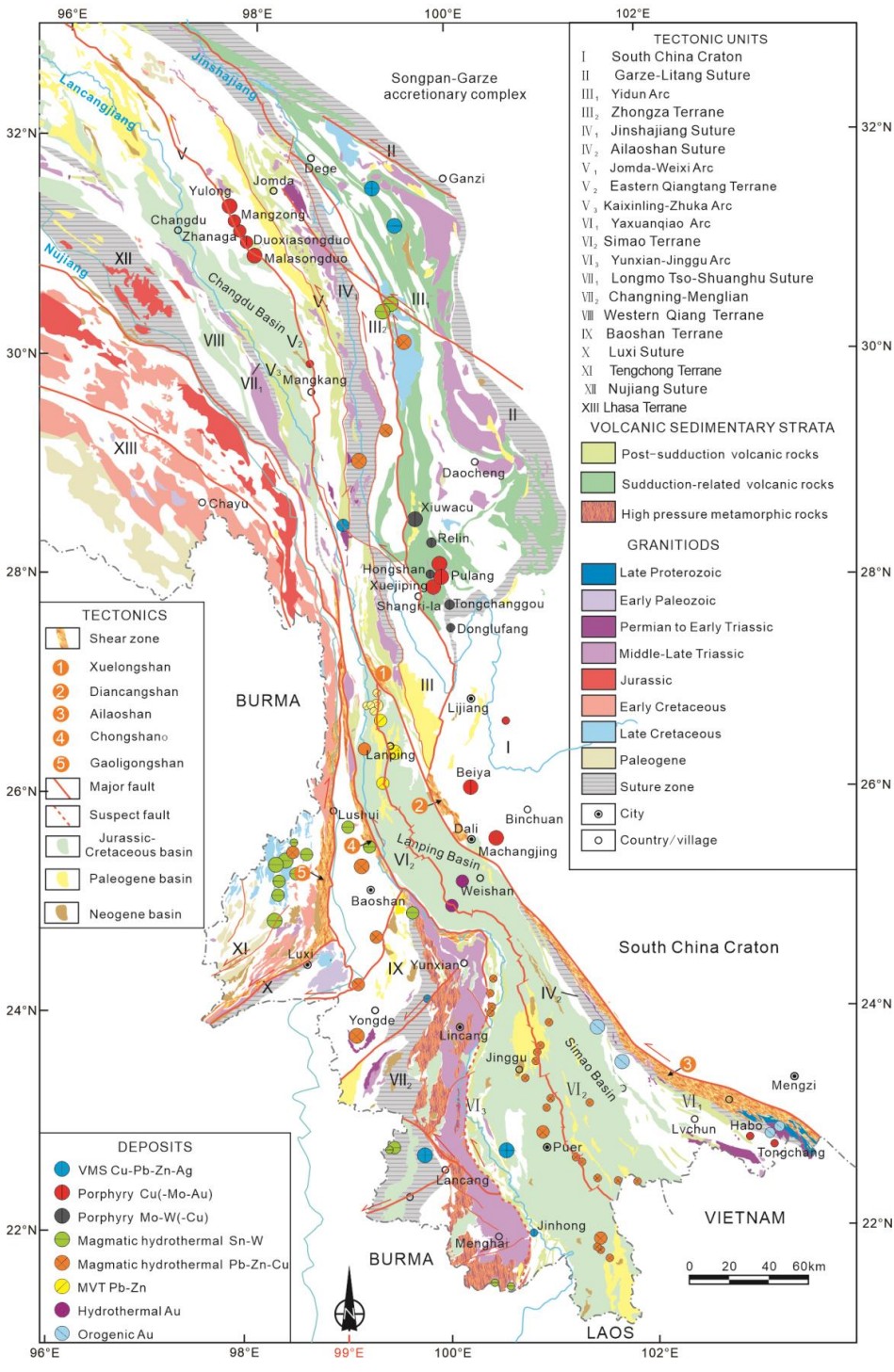

**Figure 2.** Geological map of the Sanjiang Tethyan Orogen showing the distribution of Early Paleozoic to Cenozoic igneous rocks and major ore deposit. Adapted with permission from Ref. [17]. Copyright 2016 ELSEVIER.

## 4. Methodology

Among the commonly used isotope systems, the Lu-Hf isotope system has a long half-life and is the most stable in the process of magmatic differentiation [58,59]. Therefore, Lu-Hf isotopes are used for isotopic dating and geochemical tracing. The Lu-Hf isotopic system has been rapidly developed in recent years, and the Hf isotopic signature of magmatic rocks can be used to explore the crustal nature of the magma source area (from ancient or juvenile crust) [60,61]. Three hundred and fifty-nine granitic magmatic rock samples (including 4952 zircon Hf isotope analysis points) were used in combination with existing zircon U-Pb geochronological data to evaluate the crustal evolution in this region through time [38,40,47–49,51,52,62–162]. The samples are mainly neutral-acidic rocks, including quartz diorite, diorite, granite, and granite porphyry. These samples cover almost all of the Sanjiang Tethys except for the Simao block where magmatic rocks are rarely developed. Therefore, the selected samples cover all stages of magmatic activities and can better represent the properties of regional magmatic rocks in the Sanjiang Tethys orogenic belt (Figure 3).

In order to produce a data set that is compatible with the previously published Hf-isotope analyses in the Sanjiang Tethyan Orogen, a consistent method was used to recalculate all of the data, taking a median value from the range of Hf isotopic values of individual samples [17,22,24,163]. The measured $^{176}$Lu/$^{177}$Hf ratios and a $^{176}$Lu decay constant of $1.867 \times 10^{-11}$ yr$^{-1}$ were used to calculate the initial $^{176}$Hf/$^{177}$Hf ratios [164] and the chondritic data of $^{176}$Lu/$^{177}$Hf= 0.0336 and $^{176}$Hf/$^{177}$Hf =0.282785 were used to calculate the $\varepsilon$Hf(t) and $f_{Lu/Hf}$ values [165]. Single-stage model ages ($T_{DM}{}^{1}$) and two-stage model ages ($T_{DM}{}^{C}$) were calculated relative to the depleted mantle values of ($^{176}$Hf/$^{177}$Hf)$_{DM}$ = 0.28325 and ($^{176}$Lu/$^{177}$Hf) $_{DM}$ = 0.0384, a $^{176}$Lu/$^{177}$Hf ratio of 0.015 was used for the average continental crust, and "t" is taken as the crystallization age of the zircon analyzed [166]. The $\varepsilon$Hf(t), $T_{DM}$, $T_{DM}{}^{C}$, $f_{Lu/Hf}$, $f_{cc}$, $f_{s}$, and $f_{DM}$ values were calculated using the formula [164–166]:

$$\varepsilon Hf(t) = 10000 \times \{[(^{176}Hf/^{177}Hf)_S - (^{176}Lu/^{177}Hf)_S \times (e^{\lambda\lambda t} - 1)]/[(^{176}Hf/^{177}Hf)_{CHUR,0} - (^{176}Lu/^{177}Hf)_{CHUR} \times (e^{\lambda t} - 1)] - 1\}; T_{DM} = 1/\lambda \times \ln\{1 + [(^{176}Hf/^{177}Hf)_S - (^{176}Hf/^{177}Hf)_{DM}]/[(^{176}Lu/^{177}Hf)_S - (^{176}Lu/^{177}Hf)_{DM}]\}; T_{DM}{}^{C} = T_{DM} - (T_{DM} - t) \times [(f_{cc} - f_s)/(f_{cc} - f_{DM})] f_{Lu/Hf} = (^{176}Lu/^{177}Hf)_S/(^{176}Lu/^{177}Hf)_{CHUR} - 1,$$

The Hf contour maps were produced using the inverse distance weighted interpolation methods in the MapGis program to contour the Hf dataset. In order to produce the most robust spatial representation of the isotopic dataset, this method used 12 nearest neighbors at a "power" [16,163].

Since the same sample often has multiple Hf isotope test values, we used the median for a range of Hf isotope values from an individual sample, which helped to exclude abnormal data [17,22,24,163]. The median obtained from the calculation after excluding abnormal data is not affected by the two extremes of maximum and minimum data and can represent the concentrated trend of a set of data, so it can truly reflect the data characteristics of the samples.

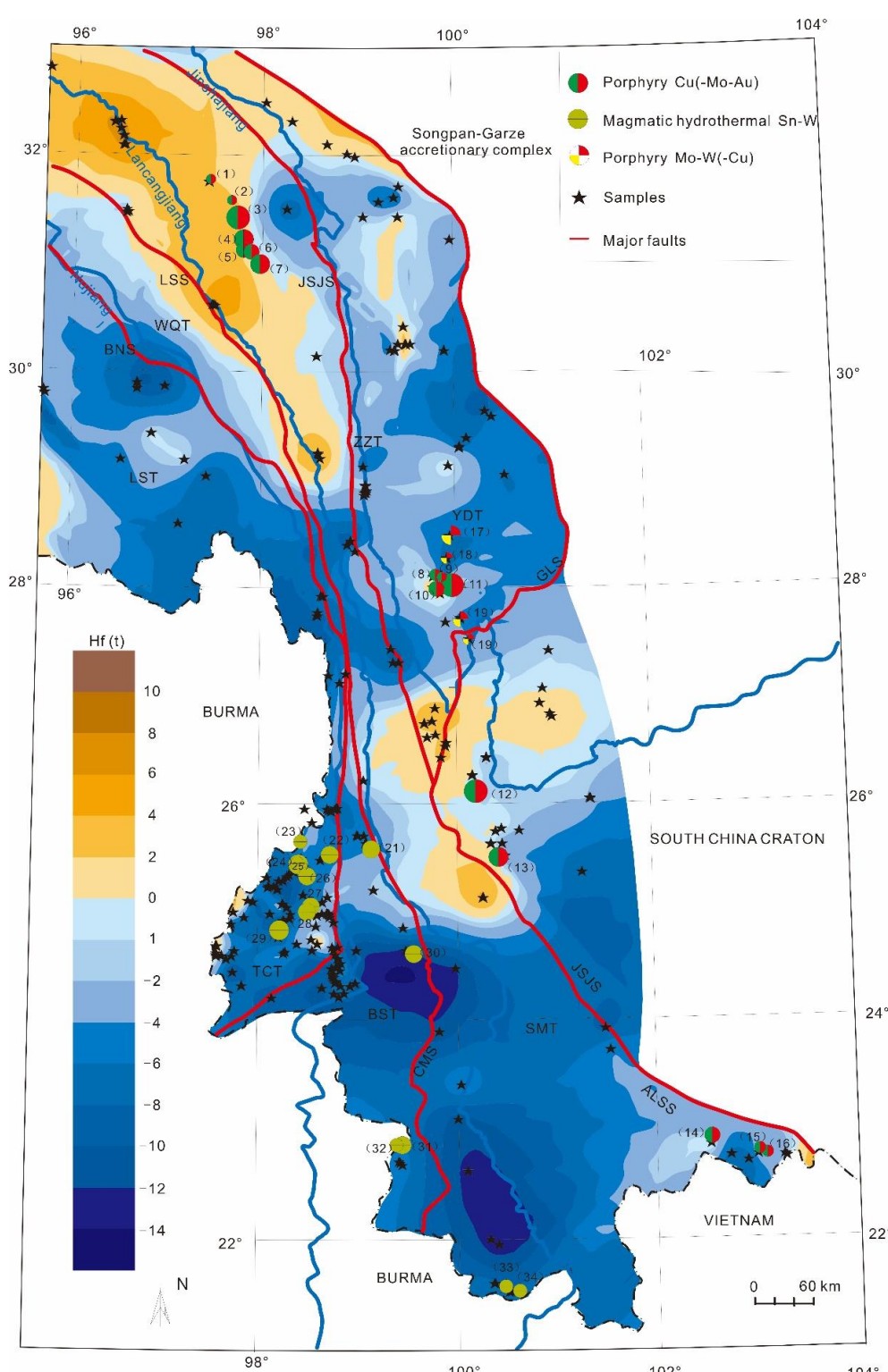

**Figure 3.** Hf isotopic contour maps showing the spatial variation of zircon εHf values for the Early Paleozoic to Cenozoic igneous rocks in the Sanjiang Tethyan Orogen. Abbreviations: BST = Baoshan terrane; EQT = Eastern Qiangtang terrane; LST = Lhasa terrane; SMT = Simao terrane; TCT = Tengchong terrane; WQT = Western Qiangtang terrane; YDT = Yidun arc terrane; ZZT = Zhongza terrane; ALSS = Ailaoshan Suture; BNS = Bangonghu–Nujiang Suture; CMS = Changning–Menglian Suture; JSJS = Jinshajiang Suture; GLS = Garzê–Litang Suture; LSS = Longmucuo–Shuanghu Suture.

## 5. Results

According to the statistics of zircon $\varepsilon$Hf(t) values and the two-stage model age of intermediate acid rocks in Sanjiang Tethys orogenic belt, the $\varepsilon$Hf(t) values range from $-14$ to $+10$, concentrated in the range of $-12$ to $+8$. The corresponding two-stage model ages range from 0.5 Ga to 2.2 Ga. Zircon $\varepsilon$Hf (t) value contours are shown in Figure 3, where the warm color system is for positive $\varepsilon$Hf (t) values, and the darker the color the larger the positive value. The cold color system is for negative $\varepsilon$Hf (t) values, and the darker the color the smaller the negative value.

The contour results show that there are several anomalous regions with high $\varepsilon$Hf(t) values in the Sanjiang Tethys orogenic belt (Figure 3). The anomalous areas with high $\varepsilon$Hf(t) values are distributed roughly along the Jinshajiang–Ailaoshan suture on both sides and mainly occur in the East Qiangtang block that belongs to the northern part of the suture ($\varepsilon$Hf(t) = 3.3, $T_{DM}{}^c$ = 0.9 Ma), the Yidun Island arc that belongs to the central part of the suture ($\varepsilon$Hf(t) = 1.1; $T_{DM}{}^c$ = 1.2 Ga), a small part of the Simao block that belongs to the southern part of the suture ($\varepsilon$Hf(t) = 4; $T_{DM}{}^c$ = 0.8 Ga), and the southern part of the South China Craton ($\varepsilon$Hf(t) = 4; $T_{DM}{}^c$ = 1.0 Ga). High $\varepsilon$Hf values are also present at the eastern margin of the Lhasa block, the eastern margin of the Tengchong block, and the southern margin of the Ailaoshan suture on a local scale.

The contour results also show several low $\varepsilon$Hf value anomalous regions (Figure 3). The low $\varepsilon$Hf anomalies are located in the Tengchong–Baoshan block ($\varepsilon$Hf(t) = $-6.5$), both sides of the Changning–Menglian suture ($\varepsilon$Hf(t) = $-11.4$), and the Zhongzhan block ($\varepsilon$Hf(t) = $-6.5$). It is worth noting that the Simao block magmatic rocks are shown as negative $\varepsilon$Hf(t) anomalies in the figure due to the lack of sufficient research data, which are influenced by the surrounding values.

Meanwhile, the contour results also show that all porphyry Cu(-Mo-Au) deposits (e.g., Yulong porphyry ore field (No. 1–7) in the northern segment, the Beiya (No. 12) and Machangqing (No. 13) ore deposits in the central, and the Habo (No. 14) and Tongchang (No. 16) ore deposits in the southern) are located in the high $\varepsilon$Hf(t) region($\varepsilon$Hf(t) > 0). Porphyry Mo-W(-Cu) deposits(e.g., Xiuwacu (No. 17), Relin (No. 18), Donglufang (No. 20)) and Magmatic hydrothermal Sn-W deposits (e.g., Xiaolonghe (No. 24) and Lailishan (No. 29)) are strictly restricted to the low $\varepsilon$Hf(t) region ($\varepsilon$Hf(t) < 0) (Figures 3 and 4, Table 1).

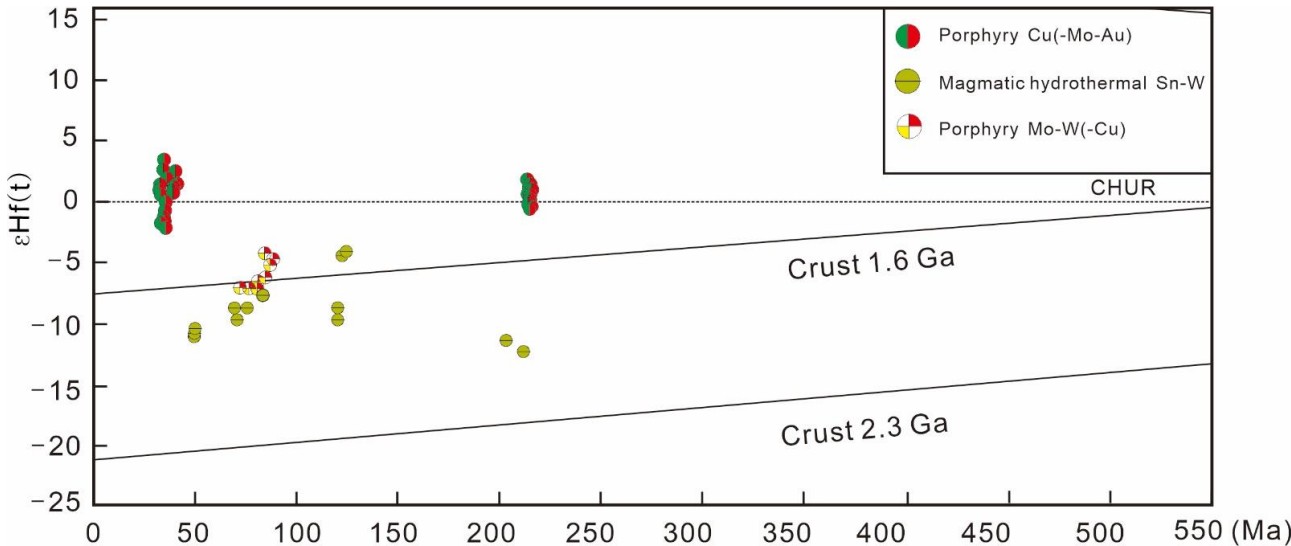

**Figure 4.** Plots of median $\varepsilon$Hf(t) versus U-Pb ages of magmatic zircons.

**Table 1.** A summary of the geological characteristics of major ore deposits in the Sanjiang Tethyan Orogen.

| Deposit | Long./Lat. | Type | Mineral Elemen | Tonnage | Grade (%) | Host Rock | Alteration | Igneous Age (Ma) | Mineralization Age (Ma) Molybdenite Re-Os | εHf (t) | Data Source |
|---|---|---|---|---|---|---|---|---|---|---|---|
| Baomai (1) | 97.43/ 31.77 | Porphyry | Cu-Mo | Cu:0.21 Mt Mo:0.06 Mt | Cu:0.22% Mo:0.06% | Biotite granite, biotite monzogranite | Potassic, propylitic, phyllic, argillic and skar | LA-ICP-MS zircon U-Pb: 37.8 ± 0.2; 42.7 ± 0.2 | 42.6 ± 0.3 | −0.4~4.9 | [47] |
| Hengxingcuo (2) | 97.71/ 31.51 | Porphyry | Cu-Mo | No data | No data | granite prophyry; | No data | No data | No data | No data | [167] |
| Yulong (3) | 97.72/ 31.42 | Porphyry | Cu-Mo-Au | Cu:6.22 Mt Mo:0.06 Mt | Cu:0.99% Mo:0.28% Au:0.35 g/t | Biotite monzogranite, granodiorite, alkali-feldspar granite | Potassic, propylitic, phyllic, argillicand skarn | LA-ICP-MS zircon U-Pb: 41.2 ± 0.2; SHRIMP zircon U–Pb: 40.9 ± 0.1 | 40.1 ± 1.8; 41.6 ± 1.4 | −0.2~4.3 | [46,47,153,168, 169] |
| Zhanaga (4) | 97.79/ 31.23 | Porphyry | Cu-Mo-Au | Cu:0.3 Mt | Cu:0.36% Mo:0.03% Au:0.03 g/t | Monzogranite, syengranite | Potassic, phyllic, argillic, propylitic | SHRIMP zircon U-Pb: 38.5 ± 0.2 | No data | 1.5~4.3 | [46,168,169] |
| Mangzong (5) | 97.79/ 31.13 | Porphyry | Cu-Mo-Au | Cu:0.25 Mt | Cu:0.34% Mo:0.03% Au:0.02 g/t | Monzogranite | Potassic, phyllic, propylitic | LA-ICP-MS zircon U-Pb: 37.6 ± 0.2 | No data | No data | [46,168,169] |
| Duoxiasongduo (6) | 97.88/ 31.11 | Porphyry | Cu-Mo-Au | Cu:0.5 Mt | Cu:0.38% Mo:0.04% Au:0.05 g/t | Monzogranite, granite, syengranite | potassic, phyllic, propylitic | SHRIMP zircon U-Pb: 37.5 ± 0.2 | 36 ± 0.4 | No data | [46,168,169] |
| Malasongduo (7) | 97.97/ 30.99 | Porphyry | Cu-Mo-Au | Cu:100 Mt | Cu:0.44% Mo:0.14% Au:0.06 g/t | Alkali-feldspar granite | Potassic, phyllic and argillic | LA-ICP-MS zircon U-Pb: 36.9 ± 0.4 | 35.8 ± 0.4 | No data | [46,168,169] |
| Lannitang (8) | 99.82/ 28.14 | Porphyry | Cu-Au | Cu:36 Mt | Cu:0.50%, Au:0.45 g/t | Quartz monzonite porphyry Diorite porphyrite | Potassic alteration, quartz sericite propylitization and argillization | LA-ICP-MS zircon U-Pb: 225.2 ± 3.5 | No data | −1.1~0.6 | [55,122,170] |
| Honhshan (9) | 99.88/ 28.12 | Porphyry | Cu-Mo | Cu:65 Mt Mo:0.58 Mt | Cu:1.23% Mo:0.14% | Quartz monzonite porphyry | Skarnization | LA-ICPMS zircon U-Pb: 75.8 ± 1.3 | 77.9 ± 1.1; 81.05 ± 1.17 | −8.8~−5.5 | [72,73,122,156] |
| Xuejiping (10) | 99.83/ 28.01 | Porphyry | Cu-Au | Cu:54.15 Mt | Cu:0.53% Au:0.06 g/t | Dioritic porphyry and monzoniticry | Potassic alterationand quartz sericit | SIMS zircon U-Pb: 218.3 ± 1.6; 218.5 ± 1.6, | 221.4 ± 1.3 | −2.7~4.4 | [55,82,100,122] |
| Pulang (11) | 99.99/ 28.04 | Porphyry | Cu-Mo-Au | Cu:1625 Mt; Mo:84.8 Kt; Au:28.8 t | Cu:0.52%; Mo:0.004%; Au:0.18 g/t | Quartz diorite porphyry, quartz monzonite porphyry | Potassic alteration silification | MS zircon U-Pb: 221.0 ± 1.0; 211.8 ± 0.5, | 218 ± 3.4; 219.7 ± 3.4 | −2.4~2 | [55,122,159,171] |
| Beiya (12) | 100.22/ 26.14 | Porphyry | Cu-Au-Fe | Cu:0.26 Mt; Au:127 t; Fe:30 Mt | Cu:0.5%; Au:2.45 g/t; Fe:35% | Quartz–albite porphyry, quartz porphyry, biotite–K-feldspar porphyry, | Potassic alteration, silicification, sericitization, chloritization, carbonization | LA-ICP-MS zircon U-Pb: 34.72 ± 0.94; 37.67 ± 0.97 | 36.82 ± 0.5 | −4.5~4.3 | [48,49] |

Table 1. *Cont.*

| Deposit | Long./Lat. | Type | Mineral Elemen | Tonnage | Grade (%) | Host Rock | Alteration | Igneous Age (Ma) | Mineralization Age (Ma) Molybdenite Re-Os | εHf (t) | Data Source |
|---|---|---|---|---|---|---|---|---|---|---|---|
| Machangqing (13) | 100.44/ 25.53 | Porphyry | Cu-Mo-Au | Cu:81,258 t; Mo:44,525 t; Au:26 t | Cu:0.50%; Mo:0.08%; Au:4.01–8.70 g/t | Syenite porphyry; monzonite porphyry; granite prophyry; limestone and sandstone | Potassic alteration, silicification, sericitization chloritization, | LA-ICP-MS zircon U-Pb: 33.78 ± 0.21; 35.6 ± 0.3; 35.0 ± 0.2; 37.93 ± 0.82 | 35.3 ± 0.7; 35.8 ± 1.6; 33.9 ± 1.1; 34.72 ± 0.5 | −2.4~1.2 | [50,172] |
| Habo (14) | 102.54/ 22.94 | Porphyry | Cu-Mo-Au | Cu:0.53 Mt Mo:37718 t | Cu:0.42– 1%Mo:0.01–0.1% Au:1–33 g/t | Biotite quartz monzogranite; quartz monzonite porphyry; monzonite porphyry | K-silicate, quartze-sericite, propylitic | LA-ICP-MS zircon U-Pb: 36.34 ± 0.63; 35.99 ± 0.36 | 35.47 ± 0.2 | −4.3~−1.1 | [51] |
| Chang'anchong (15) | 103.01/ 22.81 | Porphyry | Cu-Mo-Au | Cu:29,337 t Mo:13,310 t | Cu:1.48% Mo:0.13% Au:0.25 g/t | Quartz monzonite granite | K-silicate, quartz-sericite, skarn | LA-ICP-MS zircon U-Pb: 36.3 ± 0.3; 33.7 ± 0.8 | 34.54 ± 0.7 | No data | [172] |
| Tongchang (16) | 103.05/ 22.79 | Porphyry | Cu-Mo-Au | Cu:0.01 Mt Mo:0.02 Mt | Cu:1.24% Mo:0.218% Au:0.13 g/t | Quartz syenite porphyry | K-silicate, quartz-sericite, skarn | LA-ICP-MS zircon U-Pb: 34.6 ± 0.2; 35.1 ± 0.3 | 34.38 ± 0.5; 34.2 ± 0.3 | −4.4~−1.2 | [172] |
| Xiuwacu (17) | 99.99/ 28.5 | Porphyry | Mo-W | Mo:1.36 Mt WO$_3$:0.84 Mt | Mo:0.38% WO$_3$:0.28% | Biotite monzogranite, alkali-feldspar granite | Propylitization, potassic alteration and serictization | LA-ICPMS zircon U-Pb: 85.6 ± 0.5; 84.8 ± 0.6; 84.4 ± 1.4 | 82.3 ± 1.1; 83.5 ± 0.3 | −7.1~−3.6 | [55,72,73] |
| Relin (18) | 99.94/ 28.29 | Porphyry | Mo-Cu | | Mo:0.049% | Monzogranite, granite porphyry | Propylitization, potassic alteration and skarnization | LA-ICPMS zircon U-Pb: 82.7 ± 0.5 | 80.3 ± 1.1; 82.9 ± 1.1 | −9.0~−4.6 | [55,72,73] |
| Tongchanggou (19) | 100.08/ 27.23 | Porphyry–skarn | Mo-Cu | Mo:30 Mt Cu:0.34 Mt | Mo:0.3% Cu:0.8% | Granodiorite porphyry | | LA-ICP-MS zircon U-Pb: 85.7 ± 0.5; 84.7 ± 0.6 | 86.8 ± 0.6; 85.2 ± 0.4 | −5.7~−2.7 | [73,157] |
| Donglufang (20) | 100.16/ 27.54 | Porphyry | Mo-Cu | 80 Mt | Mo:0.15% Cu:0.48% | granodiorite porphyry | Potassic, propylitic, phyllic, argillic and skarn | LA-ICP-MS zircon U-Pb: 85.1 ± 0.5; 84.4.1 ± 0.3 | 84.9 ± 1.0 | −9.9~−0.5 | [143] |
| Tiechang (21) | 99.15/ 25.58 | Hydrothermal type | Sn–W | No data | Sn:1.22% | granite, gneiss | Sericitization, silicification, chloritization | No data | No data | No data | [17,57] |
| Dadongchang (22) | 98.73/ 25.53 | Hydrothermal type | Sn | Sn:10,000 Mt | Sn:0.14%; Pb:8.18%; Zn:18.0%; Cu:0.73 | biotite granite, dolomitic, limestone, arenaceous mudstone and sandstone | Skarnization, silicification, tremolitization, sericitization, chloritization, fluoritization | Early Cretaceous | 118.0 ± 2.4 | No data | [17,57] |

**Table 1.** *Cont.*

| Deposit | Long./Lat. | Type | Mineral Elemen | Tonnage | Grade (%) | Host Rock | Alteration | Igneous Age (Ma) | Mineralization Age (Ma) Molybdenite Re-Os | εHf (t) | Data Source |
|---|---|---|---|---|---|---|---|---|---|---|---|
| Diantan (23) | 98.43/ 25.65 | Hydrothermal type | Sn–Fe | No data | No data | alkali feldspar granite, dolomitic limestone, mudstone and sandstone | Skarnization, silicification, sericitization, chloritization | LA-ICP-MS zircon U-Pb: 122.0 ± 2.1; 123.0 ± 1.4 | No data | −3.9~−4.0 | [17,57,106] |
| Xiaolonghe (24) | 98.41/ 25.44 | Hydrothermal type | Sn | Sn:26,200 | Sn:0.18–0.42% | biotite granite, sandy slate | Greisenization, sericitization, silicification, chloritization | LA-ICP-MS zircon U-Pb: 75.2 ± 4.2; 70.5 ± 3.4 | No data | −11.3~−1.7 | [17,87,106,115] |
| Dasongpo (25) | 98.4/ 25.45 | Hydrothermal type | Sn | Sn:>1000 | Sn:1.38% | biotite granite and monzogranite, sandy slate | Greisenization, sericitization, silicification, chloritization | LA-ICP-MS zircon U-Pb: 70.3 ± 3.2; 71.5 ± 2.1 | No data | −8.4~−5.0 | [17,106] |
| Gudong (26) | 98.5/ 25.33 | Hydrothermal type | Sn | No data | No data | biotite granite, sandy slate | Greisenization, sericitization, silicification, chloritization | No data | No data | No data | [17] |
| Baihuanao (27) | 98.54/25.05 | Hydrothermal type | Sn | Sn:12,638 | Sn:0.014% | biotite albite granite | Albitization, greisenization, amazonitization | LA-ICP-MS zircon U-Pb: 61.9 ± 1.4 | No data | −11.8~−8.5 | [17,87] |
| Xinqi (28) | 98.50/ 25.00 | Hydrothermal type | Sn | No data | Sn:0.47% | monzogranite, schist, leptynite | Greisenization, silicification, sericitization, | LA-ICP-MS zircon U-Pb: 61.9 ± 1.4 | No data | No data | [17] |
| Lailishan (29) | 98.22/ 24.83 | Hydrothermal type | Sn | Sn:42,600 | Sn:1.11% | biotite granite, monzogranite, and feldspar granite; sandstone and slate | Potassic alteration kaolinization, sericitization, silicification, chloritization | LA-ICP-MS zircon U-Pb: 45.77 ± 0.89; 50.0 ± 1.6 | No data | −11.9~−8.0 | [17,93,115] |
| Haobadi (30) | 99.58/ 24.61 | Hydrothermal type | Sn | Sn:8600 | Sn:0.95% | monzogranite, sandstone, quartz sandstone, quartzite | Silicification | LA-ICP-MS Zircon U-Pb: 231.5 ± 3.6 | No data | No data | [17,57] |
| A'mo (31) | 99.46/ 22.84 | Hydrothermal type | Sn | No data | Sn:1.07% | biotite granite, two-mica granite, and pegmatite, marble, schist | Albitization | No data | Taeniolite Rb-Sr age, 21.5 | No data | [17,57] |
| Damasa (32) | 99.40/ 22.84 | Hydrothermal type | Sn | No data | No data | biotite granite | No data | No data | No data | No data | [17,57] |
| Bulangshan (33) | 100.45/ 21.43 | Hydrothermal type | Sn | No data | No data | granite | Albitization, tourmalinization, silicification, greisenization | LA-ICP-MS zircon U-Pb: 216 ± 1; 218 ± 1 | No data | −10.8~−7.4 | [17,57,160] |
| Mengsong (34) | 100.54/ 21.38 | Hydrothermal type | Sn | No data | No data | granite | Albitization, tourmalinization, silicification, greisenization | LA-ICP-MS zircon U-Pb: 228 ± 2; 222 ± 1, | No data | −14.4~−10.1 | [17,57,160] |

## 6. Discussion

### 6.1. Distribution of Different Deposits Constrained from Zircon Hf-Isotopic Mapping

Previous knowledge of the lithospheric structure and deep tectonic framework of the Sanjiang Tethys orogenic belt was mainly speculated by geophysical means [37,173] but lacked material information support. Neutral-acid rocks are widely developed in the Sanjiang Tethys orogenic belt. These zircon Hf isotopes can be used as a "rock probe" to trace the petrogenesis, so as to reflect the material composition and temporal and spatial changes in different parts of the crust. In this study, based on the contour map of granitic zircon $\varepsilon$Hf(t) values, we can demonstrate the spatial distribution and spatial transformation of the model ages of deep source rocks in the crust, and thus determine the spatial distribution and temporal evolution of different terranes.

Zircon Hf isotope mapping shows that there are widespread negative $\varepsilon$Hf(t) values in the Tengchong block, Baoshan block, and Changning–Menglian suture in the west of Sanjiang Tethys orogenic belt and Zhongzan block Yidun island arc belt in the east, which correspond to two-stage model ages $T_{DM}{}^c > 1.2$ Ga, suggesting that these terranes may be ancient, locally modified crustal blocks. Spatially, the higher $\varepsilon$Hf(t) values correspond to a younger $T_{DM}{}^c$ age in the east Qiangtang block, the southern part of the Yidun Island arc, the Simao block, and the southern margin of the South China Craton, which are located in the eastern part of the Sanjiang Tethys orogenic belt, indicating that the petrogenesis of these regions is dominated by newly formed mantle-derived components.

Combining Hf isotope mapping results with geophysical exploration, we can construct a three-dimensional lithospheric structure at the scale of the terrane. Zhou [37] speculated that the crust of the southwest Sanjiang Tethys orogenic belt shows a "stepwise" thickening trend from west to east and from south to north. The 3D model map from the terrane to the South China Craton shows that the Tengchong–Baoshan block has an approximately 35 km thick crust with negative $\varepsilon$Hf(t) values and old model ages ($T_{DM}{}^c$ values), indicating the presence of older crustal components and remelted crustal components in the magmatic rocks of the region [17,34]. In contrast, the continental crust of the eastern South China Craton is about 45 km thick and the magmatic zircons have positive $\varepsilon$Hf(t) values and young model ages ($T_{DM}{}^c$ values), indicating the incorporation of juvenile mantle-derived material in the magma source area of the region [17,34].

Therefore, based on zircon Hf isotope regional mapping, the Sanjiang region shows temporal and spatial heterogeneity of the crust, which is bounded by the Longmucuo–Shuanghu suture and the Changning–Menglian suture. The ancient crust of negative $\varepsilon$Hf(t) values is mainly in the west and the juvenile crust of positive $\varepsilon$Hf(t) values is mainly in the east.

The zircon Hf isotope mapping of granitic rocks in the Sanjiang region reveals that the mineralization system in the Sanjiang region is controlled by the crustal properties in the region, and the differences in crustal properties constrain the distribution pattern of different mineralizing elements (Figure 3). The results of zircon Hf isotope mapping prove that the properties and composition of the crust are closely related to the formation and spatial distribution of polymetallic deposits. Firstly, all porphyry Cu-(Mo-Au) are located within the juvenile crustal blocks with high $\varepsilon$Hf values. This is consistent with the Cu-ore-forming magmas in the Jinshajiang-Ailaoshan suture mainly sourced from the juvenile lower crust [94]. Secondly, all magmatic-hydrothermal Sn-W deposits and porphyry Mo-W(-Cu) deposits are strictly controlled by the reworking of an ancient crust with negative $\varepsilon$Hf values and are mainly distributed in the Tengchong block, the Yidun island arc, and the Changning–Menglian suture (Figures 3 and 4).

### 6.2. Exploration for the Sanjiang Tethyan Orogenic Belt

The Triassic porphyry Cu(-Mo-Au) deposits and Middle Eocene to Early Oligocene porphyry Cu(-Mo-Au) deposits are developed in a magmatic arc within the Garzê–Litang suture and the Jinshajiang–Laohuoshan suture zone, with a high $\varepsilon$Hf(t) region (Figures 3 and 4). Ore-bearing porphyry magmas forming the juvenile lower crust in the Sanjiang Tethyan orogenic belt were derived from asthenospheric mantle wedge metasomatized by the

upwelling of the asthenosphere in post-subduction and post-collision settings [17]. The underplating of these magmas at the bottom of the crust inevitably led to the enrichment of the juvenile lower crust in Cu(-Mo-Au) contents (Figure 5) [16]. These results suggest that the juvenile crust plays a major controlling role in the formation of porphyry Cu(-Mo-Au) deposits.

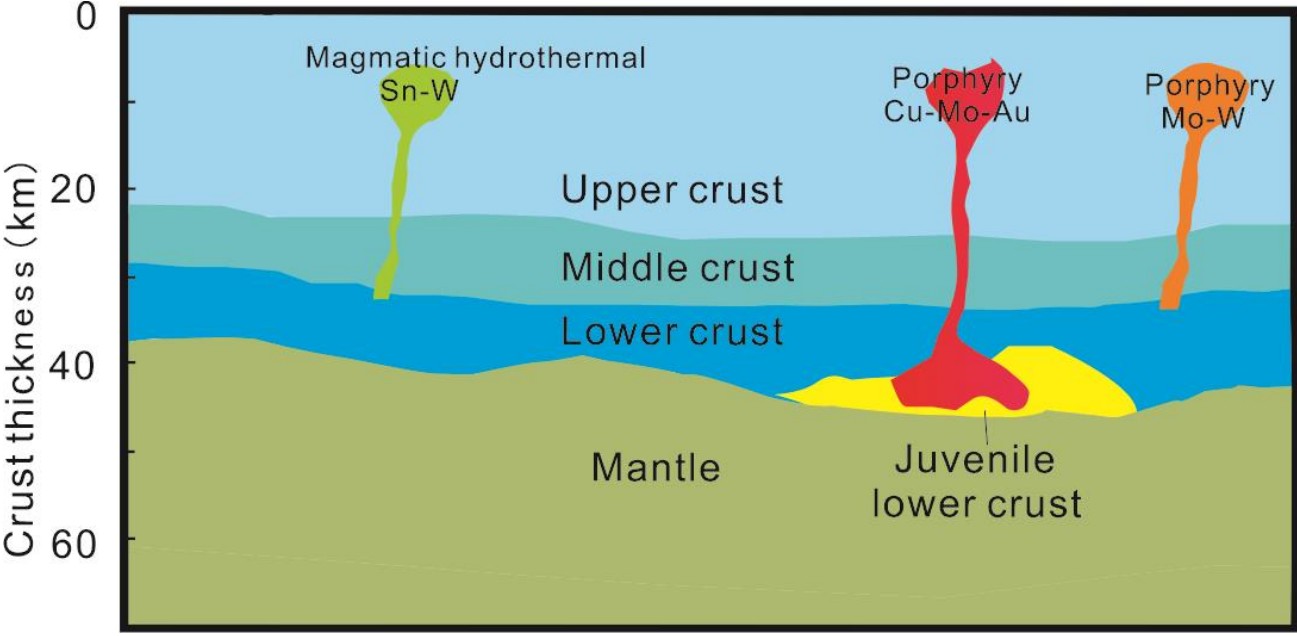

**Figure 5.** Model of composite metallogenic systems.

Late Triassic I-type granites and related porphyry Cu–Mo deposits (e.g., Xuejiping (No. 10), Pulang (No. 11)) occur in the southern part of the Yidun Arc (Figure 3 and Table 1). The porphyry Cu mineralization in the southern part of the Yidun Arc region most likely originated from the enriched mantle wedge metasomatized by subduction-derived fluids and sediments [55,174]. The subduction-derived fluids and sediments not only provided the volatile components ($H_2O$ and Cl) but also controlled the high oxygen fugacity ($f_{O2}$), which makes calc-alkaline magmas favorable for porphyry Cu mineralization [55,174].

A small area of high $\varepsilon Hf(t)$ anomaly is distributed in the Yidun Island Arc, mainly located at the southern margin of the Yidun Arc. The area reveals medium acidic volcanic rocks of the Upper Triassic Tumgou Formation interspersed with shallow metamorphic sandy mudstone and carbonate rock stratigraphy [45,55]. In general, it is an NW-trending compound anticline structure that developed NW-trending faults [55,175]. The medium acidic (porphyritic) rocks are widely distributed in groups, which can be divided into three rock belts from east to west: the Late Triassic island arc porphyry belt in the eastern north, the Early Triassic genus ophiolitic melange belt in the middle, and the Early and Middle Triassic porphyry belts in the southwest [53]. This area is located in the North-South positive magnetic anomaly zone, and both sides are gentle negative magnetic field areas. Regional chemical prospecting has identified several "high, large, and complete" anomalies with high intensity, large lining, large scale, good concentration zoning, and obvious concentration centers, such as Cu, Mo, Pb, Zn, Ag, and Au [53]. The remote sensing image shows a dense intersection of northwest and northeast linear tectonics, forming a fine rhombic network of fracture blocks, with magma rings and hydrothermal rings developed in complex combinations of overlapping, concentric, and offset superposition [53,175]. Important porphyry Cu(-Mo-Au) deposits such as Pulang (No. 11) ultra-large porphyry, Xuejiping (No. 10), and Hongshan (No. 9) have been evaluated in this area [45,55,171,176], and its periphery, Pushang and Songnuo, have a similar metallogenic background and mineralization characteristics as Pulang. It is a favorable area for finding porphyry Cu(-Mo-Au)

deposits, and the prospecting prospect is good [53], which is consistent with the area with a high abnormal value of Hf in this paper.

Middle Eocene to Early Oligocene potassic–ultrapotassic intrusive rocks and related porphyry Cu–Mo-Au deposits dominantly extend along the Jinshajiang–Ailaoshan tectonic belt [30,33]. The potassic–ultrapotassic intrusive rocks are derived from the partial melting of a thickened, potassic, mafic lower crust with minor input from an older igneous felsic component. Partial melts of K-rich mafic lower crust produced high-K calc-alkaline granitic intrusions and porphyry Cu-Mo deposits [94].

The high $\varepsilon$Hf(t) anomaly region in the northern part of the Jinshajiang–Ailaoshan suture is between the Jinshajiang binding zone and the Nujiang binding zone. In between, the Jinshajiang basal–ultramafic zone, Baimaxueshan granite belt, Yulong alkaline granite belt, the Leiwuqi–Dongdashan granite zone, Nujiang-Bitu basic ultrabasic rock belt, Guoqingchaw granite belt, and Zaxize–Sanmiancun granite belt are developed from east to west [53]. The geochemical anomalies in this area are controlled by regional geological structure and magmatic rocks and are distributed in the NW–SE direction. From east to west, it can be clearly divided into four anomalous zones: Jinshajiang Au and Cu polymetallic anomalous zone; Yulong porphyry Cu and Mo anomalous zone; Lancangjiang W, Sn, Pb, Zn, Ag, Cu polymetallic anomalous zone; and Nujiang Au, Pb, Ag, W, Sn, Cu polymetallic anomalous zone [53]. The discovered and evaluated large and medium-sized porphyry Cu(-Mo-Au) deposits are all produced in the banded Yulong porphyry Cu and Mo anomaly, which is consistent with the high $\varepsilon$Hf(t) anomaly area. In addition, the distribution characteristics of the enrichment degree and dispersion of elements in the earth rocks and aqueous sediments of the eastern Tibetan region suggest that the eastern Tibetan earth has very good prospects for polymetallic mineralization such as Cu-Mo, Au, Pb, and Zn [53,177], which is consistent with the region of high anomalous values of Hf in this paper.

A high $\varepsilon$Hf(t) anomaly region is also seen in the south-central part of the Jinshajiang–Ailaoshan suture. This anomalous region is consistent with the distribution of the Himalayan alkali-rich porphyry belt in western Yunnan. The alkali-rich porphyry belt in western Yunnan can be divided from north to south into six alkali-rich porphyry groups: the Bengge–Taohua Group, the Yongsheng–Ninglang Group, the Machangqing–Beiya–Liuhe Group, the Yongping–Weishan Group, the Yaoan–Huaping Group, and the Jinping–Luchun Group (Figure 6) [178,179]. The quartz diorite porphyry and quartz diorite porphyrites in the contact zone of the Jinsichang and Taohua areas (Figure 6) are commonly developed with strong silicification, hornification, yellow (brown) iron mineralization, and skarnization. Large-scale physical surveys in the Taohua–Jinsichang area also show the existence of concealed ore bodies and concealed rock bodies and the presence of chemical anomalies of Cu, Pb, Zn, Ag, Au, Sb, and other elements [50]. In the area of Ninglang–Yongsheng (Figure 6), 47 gold anomalies were traced by the chemical anomalies, which were distributed in the NE direction and coincided with the Cu anomalies. The Cu anomaly is accompanied by the main fracture in a band-like spreading, forming a concentration center with w (Cu) > $300 \times 10^{-6}$ in the Bainiuchang and Luobodi areas, generally containing w (Au) of $(5\sim7) \times 10^{-9}$ and w (Ag) of $(0.15\sim1.2) \times 10^{-6}$ [53]. The Beiya and Heqing (Figure 6) areas are dominated by chemical anomalies of gold, with a large scale, high intensity, and obvious concentration center. Important gold anomalies include Songgui, Tanyao, Beiya, and Huadianba. In the Tiesuodongshan area of Yongren, Yunnan, the geochemical anomalies are mainly Cu, Pb, Au, Pb, Ag, and Mo, among which the copper anomalies mostly coincide with the Cu deposits in the area, and some of the gold-copper anomalies are located in the distribution area of porphyry bodies or their sides [53]. In the Weishan–Yongping area (Figure 6), regional chemical probes show Au, Sb, Hg, As, Cu, Co, Pb, Zn, Ag, and other elemental anomalies [180]. Au anomaly higher values are mainly located in Yongping Zhuopan, Yangbi Huanglianpu, Weishan Zijinshan, Weishan Lianhuashan, and Nanrun Gonglanghu areas [180]. The higher value of Cu anomalies is mainly concentrated in Yongpingchang Street, Shuixie, and Nanjian Gonglanghu areas,

and the Co anomaly is basically similar to the Cu anomaly [180]. West Yunnan has a large distribution of Himalayan copper-bearing porphyry bodies, and porphyry alteration and strong Cu mineralization are commonly seen. It is a favorable area for exploring porphyry Cu(-Mo-Au) deposits.

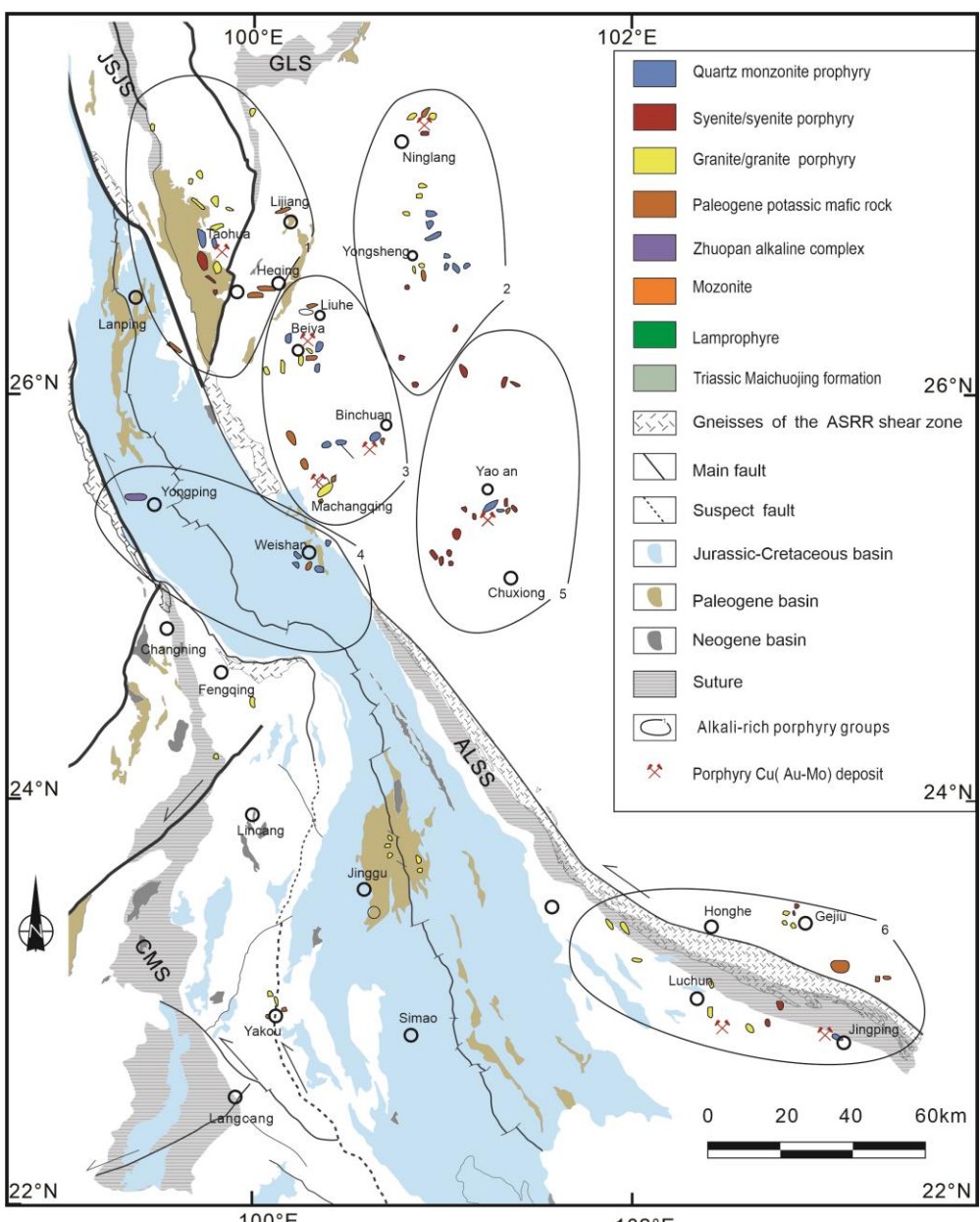

**Figure 6.** Cenozoic alkali-rich porphyry showing distribution in western Yunnan. Adapted with permission from Ref. [152]. Copyright 2020 Wiley Online Library.

The spatial distribution of porphyry Mo-W(-Cu) deposits and magmatic hydrothermal Sn-W deposits are strictly restricted to the low $\varepsilon$Hf(t) region (Figure 3). Proven porphyry Mo-W(-Cu) deposits (Xiuwacu (No. 17), Relin (No. 18), Tongchanggou (No. 19), and Donglufang (No. 20)) are concentrated at the edge of reworked old crust or near the ancient crust. Similarly, the identified Magmatic hydrothermal Sn-W deposits (Xiaolonghe (No. 24), Dasongpo (No. 25), Lailishan (No. 29)) are concentrated in the ancient $T_{DM}{}^c$ crustal zone or developed along the $\varepsilon$Hf(t) value isotopic boundary of ancient crustal blocks. These features suggest that the spatial distribution of Sn-W and Mo-W deposits is closely related

to the compositional heterogeneity and remelting/modification of ancient crustal blocks (Figure 5).

Late Cretaceous magmatic rocks are developed in Xiuwacu, Relin, Donglufang, and other areas on the southern margin of the Yidun Island Arc (Figure 2). The magmatic activity in this period is that the Yidun Island Arc is in the extensional background, due to the thinning of the lithosphere. The asthenosphere material upwelled, the temperature of the lower part of the rock group increased, and the thickened lower crust was partially melted [55]. During the partial melting of the lower crust, chalcophile elements (such as Cu) were removed, but siderophile metals (such as Mo) were left as a residue in the cumulate zone of the deep arc crust and/or in the metasomatized mantle lithosphere. This mechanism facilitated porphyry Mo-Cu mineralization in the southern Yidun Terrane (Figure 5) [156]. Elemental and chemical anomalies of Mo, W, Sn, and Bi in the Yidun Island arc region are mainly distributed in the Xiuwacu, Yaza–Lantang–Pulang, and Gelu areas of the Garzê porphyry belt [56]. Among them, the Mo, W, Sn, and Bi anomalies located in the Xiuwacu promotion site are large in scale, high in intensity, and obvious in concentration centers, with tertiary concentration zoning, representing the strong Mo-W mineralization in the area [56]. These areas of chemical anomalies are generally consistent with the transformation sites of the low $\varepsilon$Hf(t) isotope anomalies in zircon.

The western Yunnan Sn belt is mainly developed in the Teng–Liang area, Baoshan block, Changning–Menglian, and Lincang combined belt, all in the region of low zircon $\varepsilon$Hf(t) values. The Sn bearing granites in western Yunnan can be divided into three granite belts: Yunlong–Mengnan (East Asian belt), Changning–Ximeng (Central Asian belt), and Tengchong–Lianghe (West Asian belt) [181]. The comparative research revealed that the Sn-bearing granite belt in western Yunnan belongs to the northern extension of the Sn-bearing granite belt in Southeast Asia [182]. The Sn-bearing granites are produced by regional tectonics driven by micro-plate subduction and collision [115]. At present, most of the discovered deposits in the Sn belt in western Yunnan are related to the remodeling–remelting granite magmatic activity [115]. The remodeling–remelting of the crust enriches the metallogenic metal elements initially dispersed in the crust (mantle). Regional geophysical studies show that Sn polymetallic deposits in western Yunnan are mainly located in the low-gravity anomaly zone (−2 mGal~−10 mGal), and the range of weak positive aeromagnetic anomalies (0–100 nT) reflects that the Sn polymetallic deposits in western Yunnan are mainly associated with granites and sedimentary rocks [115]. Regional chemical anomalies from east to west can be divided into the Tengchong Sn polymetallic anomaly; the Gaoligongshan–Longchuan Sn, W, Be, Nb anomaly; the Luxi Sn, W, Be anomaly; and the Changning–Ximeng Sn, B anomaly [183]. Large and medium-sized deposits such as Lailishan and Xiaolonghe have been discovered in these anomalous zones.

The crustal structure constrains the formation of metallogenic systems and the spatial distribution of mineral deposits. Therefore, zircon Hf isotope mapping can be used as a method to constrain regional mineralization targets by combining regional geological features and geophysical and geochemical anomalies. The northern part of the East Qiangtang region, the central part of the Jinshajiang–Ailaoshan suture, the southern extension of the Ailaoshan suture, and the Vietnam region, which show high $\varepsilon$Hf(t) anomalies, may have prospective areas for porphyry Cu(-Mo-Au) deposits. In parts of the Tengchong–Baoshan block—that is, the anomaly with low $\varepsilon$Hf(t)—there may be promising areas of Magmatic hydrothermal Sn-W deposits. The southern edge of the Yidun Island Arc, the transformation site of the zircon $\varepsilon$Hf(t) isotope anomaly, has the potential to host prospective porphyry Mo-W(-Cu) deposits.

## 7. Conclusions

Through the Hf isotope mapping of granite-like zircon in the Sanjiang area, the properties and components of the crust in the Sanjiang area are revealed. It was concluded that the eastern part of the crust is dominated by the juvenile crust and the western part by the ancient crust. Meanwhile, from the Hf isotope mapping of granite-like zircon in



the Sanjiang area, it is deduced that the lithospheric structure and its crustal type are the first-order factors that constrain the distribution of different minerals. Zircon Hf isotope regional mapping, combined with regional geological features, geophysical anomalies, and geochemical anomalies, can be used to predict regional mineralization. The northern part of the East Qiangtang area, the central part of the Jinsha River–Ailaoshan suture, the southern extension of the Ailaoshan area, and the Vietnam area may become the most potential porphyry Cu(-Mo-Au) metallogenic areas. There are potential areas for magmatic hydrothermal Sn-W deposits in some areas of the Tengchong–Baoshan block. The southern margin of the Yidun Island Arc, where the $\varepsilon$Hf(t) isotopic anomaly is transformed, is a potential prospective area for porphyry Mo-W(-Cu) deposits. This study shows that Hf isotope mapping can reveal the regional metallogenic rules and explore metallogenic prediction and metallogenic potential evaluation. Hf isotope mapping can be an expected new direction for studying regional metallogenic regularity, including, especially, studies of the detection and metallogenic background of deep crustal material.

**Author Contributions:** Conceptualisation, B.D., L.Y. and Z.Y.; methodology, B.D. and G.L.; software, L.W. and J.L.; validation, B.D.; formal analysis, B.D. and Q.C.; investigation, B.D., Z.Y., L.Y., Q.C., J.Z., K.S. and G.L.; writing—original draft preparation, B.D., Z.Y., K.S. and G.L.; writing—review and editing, B.D. and Z.Y.; visualisation, B.D. and Z.Y.; supervision, B.D. project administration, B.D. All authors have read and agreed to the published version of the manuscript.

**Funding:** This research is jointly supported by the National Natural Science Foundation of China (Numbers 41872080, 92162101), the most Special Fund from the State Key Laboratory of Geological Processes and Mineral Resources, China University of Geosciences (Number MSFGPMR201804).

**Institutional Review Board Statement:** Not applicable.

**Informed Consent Statement:** Not applicable.

**Data Availability Statement:** Not applicable.

**Acknowledgments:** The authors thank the team members at CUGB for their field support, data analysis, constructive discussions, and comments.

**Conflicts of Interest:** The authors declare no conflict of interest.

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
