# Peer review of "Zircon Hf-Isotopic Mapping Applied to the Metal Exploration of the Sanjiang Tethyan Orogenic Belt, Southwestern China"

_applsci, doi:10.3390/app12084081_

Round 1

Reviewer 1 Report

The paper has an interesting premise that Lu-Hf isotope data have metallogenic implications. The paper, however, needs serious English editing. I have attached editorial comments on just the first page, to do so for the whole paper would be essentially co-authoring it.

Aside from the text editing, I have concerns about the data the authors have used. There aren't any in this paper, just line 141 where it's stated that "This study summarizes 359 pieces [sic]  of isotope data". There are no data tables! I can not evaluate the quality of the data and thus can not evaluate the map on Figure 2 and indeed the conclusions. What was analysed?, presumably whole rocks but how big were they? how were they collected? How were hey analysed? etc. 

Author Response

Response to Reviewer 1 Comments

Point 1: The paper has an interesting premise that Lu-Hf isotope data have metallogenic implications. The paper, however, needs serious English editing. I have attached editorial comments on just the first page, to do so for the whole paper would be essentially co-authoring it.Aside from the text editing, I have concerns about the data the authors have used. There aren't any in this paper, just line 141 where it's stated that "This study summarizes 359 pieces [sic]  of isotope data". There are no data tables! I can not evaluate the quality of the data and thus can not evaluate the map on Figure 2 and indeed the conclusions. What was analysed?, presumably whole rocks but how big were they? how were they collected? How were hey analysed? etc. 

Response:  This is an important suggestion. We re-examined the language of the manuscript and supplemented table and analysis method (See in the revised manuscript)

Reviewer 2 Report

The objective of the manuscript "Zircon Hf-isotopic mapping ...." by Bin Du and others is to map the distribution of Hf-isotopic data across the Sanjiang Tethyan Oroigenic Belt and then correlate the map data with the distribution of metal ore deposits to determine the occurrence of any consistent tectonic control. They were able to successfully achieve their objective and were able to show that high zircon Hf values correlate with porphyry-copper deposits while low values correlate with magmatic hydrothermal tin-tungsten deposits.  They point out that Hf-isotopes are able to distinguish between ancient crust and juvenile crust and that porphyry-copper deposits are mainly found in juvenile crust while tin-tungsten deposits are found mainly in ancient crust.  Similar results were obtained by Hou et al. [16] and Wang et al . [23] so the concept is not particularly original, but Bin Du and others were able to successfully extend the technique to a large area of western China.  The only disappointment is the lack of any insight into the reasons why high zircon Hf-isotopic values correlate with juvenile crust and porphyry-deposits and why low values correlate with ancient crust and tin-tungsten deposits.  A few brief attempts are made but they are quite superficial.  For examples:

Lines 280 -285 "In addition, the distribution characteristics of the enrichment degree and dispersion of elements in the earth rocks and aqueous sediments of the eastern Tibetan region suggest that the eastern Tibetan earth has very good prospects ..."  So, what is it, in addition to their high or low Hf isotope content, about the earth rocks and aqueous sediments and Tibetan earth that makes them so likely to be mineralized ?

Lines 323-324 "... the spatial distribution of Sn-W and Mo-W deposits is closely related to the compositional heterogeneity and remelting/modification of ancient crustal blocks." Please explain more about how compositional heterogeneity and remelting/modification can influence the Sn, W, and Mo content of the ancient crustal blocks

Lines 345-346.  The tin deposits are "related to the transformation-remelting granite-magmatic activity."  A few more details are provided but the relationship of metal concentrations to transformation-remelting granite is not made clear.

Another weakness of the manuscript is the overuse of lists of geographic locations.  It makes the manuscript difficult to read and quite confusing.  If, perhaps some of the locations were grouped together it would be more readable. 

Finally, I suspect few readers outside of China will know where the Sanjiang Tethyan Orogenic Belt is located, so it will be important to add the location "China" to the title.

Author Response

Response to Reviewer 2 Comments

Point 1: The only disappointment is the lack of any insight into the reasons why high zircon Hf-isotopic values correlate with juvenile crust and porphyry-deposits and why low values correlate with ancient crust and tin-tungsten deposits. 

Response 1: The manuscript has been thoroughly rewritten.

Point 2: Lines 280 -285 "In addition, the distribution characteristics of the enrichment degree and dispersion of elements in the earth rocks and aqueous sediments of the eastern Tibetan region suggest that the eastern Tibetan earth has very good prospects ..."  So, what is it, in addition to their high or low Hf isotope content, about the earth rocks and aqueous sediments and Tibetan earth that makes them so likely to be mineralized ?

Response 2:  We have added reference and descriptions (See line 270-285 and line 308-313 in the revised manuscript)

Point 3: Lines 323-324 "... the spatial distribution of Sn-W and Mo-W deposits is closely related to the compositional heterogeneity and remelting/modification of ancient crustal blocks." Please explain more about how compositional heterogeneity and remelting/modification can influence the Sn, W, and Mo content of the ancient crustal blocks.

Response 3: Responds: The manuscript has been thoroughly rewritten. (See line 378-381 in the revised manuscript)

Point 4: Lines 345-346.  The tin deposits are "related to the transformation-remelting granite-magmatic activity."  A few more details are provided but the relationship of metal concentrations to transformation-remelting granite is not made clear.

Response 4: We have added reference and descriptions (See line 401-405 in the revised manuscript).

Point 5: Another weakness of the manuscript is the overuse of lists of geographic locations.  It makes the manuscript difficult to read and quite confusing.  If, perhaps some of the locations were grouped together it would be more readable.

Response 5: We have revised Figure 2 and added Table 1 and Figure 4 to help the reader understand geographic locations.

Point 6: I suspect few readers outside of China will know where the Sanjiang Tethyan Orogenic Belt is located, so it will be important to add the location "China" to the title.

Response 6 : The title has been changed to“Zircon Hf-isotopic mapping as a new method for exploration: A case study of the Sanjiang Tethyan Orogenic Belt,in Southwest China”

Reviewer 3 Report

It would be interesting for the reader to be able to download a GIS-map file (or other) in electronic form, which combines data on zircon Hf isotope, geology, regional geological features, geophysical anomalies and geochemical anomalies.

Author Response

Response to Reviewer 3 Comments

Point : It would be interesting for the reader to be able to download a GIS-map file (or other) in electronic form, which combines data on zircon Hf isotope, geology, regional geological features, geophysical anomalies and geochemical anomalies

Response : Thank you for your comments.

Reviewer 4 Report

The manuscript is devoted to the application of hafnium isotopy of zircons for metallogenic analysis and prediction. The method is new, interesting and important, especially for regions with insufficient exposure - forest cover, development of agricultural land and weathering crusts. However, the work leaves a double impression due to a number of shortcomings.

  1. The methodology section is unsatisfactory. It is not clear how the samples were taken, how the zircons were extracted, how, where, on what and by whom they were measured.
  2. Too long paragraphs, difficult to read, especially in sections 2 and 5.2.
  3. The use of abbreviations - if VMS is more or less known, then MVT encounters difficulties in understanding. It is desirable to give in full and the abbreviation in brackets as a further designation.
  4. The presentation of the results took 3 small paragraphs and half a page.
  5. in Fig. 2 shows that 8 samples were taken from Vietnam, but the results are not shown. In addition, fanciful isolines are not supported by sampling points.
  6. There are only 2 illustrations in the article - no photographs of samples, zircon grains, graphs, histograms.

Author Response

Response to Reviewer 4 Comments

Point 1: The methodology section is unsatisfactory. It is not clear how the samples were taken, how the zircons were extracted, how, where, on what and by whom they were measured.

Response 1:  According to the reviewer’s suggestion, we have rewritten the methods section (See line 150-176 in the revised manuscript).

Point 2: Too long paragraphs, difficult to read, especially in sections 2 and 5.2.

Response 2: Thank you for your comments. We have split the second part into 2 parts: Hf isotope mapping’s application to mineral prospection and Geologic Setting (See lines 59 and 93 in the revised manuscript). So the 5.2 section becomes 6.2 section.  The 6.2 section simplified.

Point 3: The use of abbreviations - if VMS is more or less known, then MVT encounters difficulties in understanding. It is desirable to give in full and the abbreviation in brackets as a further designation.

Response 3: Corrected as suggested (see lines 110 and 112 of the revised).

Point 4 The presentation of the results took 3 small paragraphs and half a page.

Response 4: We have added (see lines 201-207 of the revised).

Point 5: in Fig. 2 shows that 8 samples were taken from Vietnam, but the results are not shown. In addition, fanciful isolines are not supported by sampling points

Response 5: We have removed samples were taken from Vietnam in Fig. 2.

Point 6: There are only 2 illustrations in the article - no photographs of samples, zircon grains, graphs, histograms.

Response 6: Responds: Thank you for your comments. We have revised Figure 2 and added Table 1 and Figure 4 to help the reader understand geographic locations . No data from self-tests are available for this paper, so no photographs of samples, zircon grains.

Round 2

Reviewer 1 Report

The paper contains a significant compilation of deposits, their age dates and other relevant data. There is nothing comparable for the Hf isotope data which should be present in table form. The paper needs a regional inset map for Figure 1 showing the location of the figure. The paper needs to edited for English usage, in particular the grammatical particle "the" which is misused throughout (it's either not where it should be, or is where it shouldn't be). I would like to help with editing but do not have the time. Sections like line 138-139 are typical "In order to accurately obtain the structural characteristics of .......... than 280 samples from ref. [17]" is  not a sentence. Line 140 "pieces" of data???

Author Response

Point 1: The paper contains a significant compilation of deposits, their age dates and other relevant data. There is nothing comparable for the Hf isotope data which should be present in table form.

Response: Thank you for your comments. We have supplemented the Hf isotope datas in table1 (See in the revised manuscript)

Point 2 The paper needs a regional inset map for Figure 1 showing the location of the figure.

Response: Thank you for your comments.We have added a regional map(See in the revised manuscript).

Point 3:The paper needs to edited for English usage, in particular the grammatical particle "the" which is misused throughout (it's either not where it should be, or is where it shouldn't be). I would like to help with editing but do not have the time. Sections like line 138-139 are typical "In order to accurately obtain the structural characteristics of .......... than 280 samples from ref. [17]" is  not a sentence. Line 140 "pieces" of data???

Response: Thank you for your comments. We re-examined the language of the manuscript  (See in the revised manuscript)

Reviewer 2 Report

The revised manuscript "Zircon Hf-isotopic mapping as a new method for exploration: A case study ...." by Bin Du and others is an improvement over the earlier draft.  Most of my questions were answered appropriately although I would still like to have a better answer as to why it is that high zircon Hf isotopic values correlate with juvenile crust and porphyry-copper deposits while low negative values correlate with ancient crust and magmatic hydrothermal tin-tungsten deposits.  Lines 90 to 92 are not enough.  A more in-depth explanation is needed.  Why is the Hf-isotopic content and the base-metal content of juvenile crust so different from ancient crust?  Figure 2 is excellent and worthy of publication but the geochemical basis needs clarifying.  I am also still concerned that the previously published work by Hou et al. [16], Wang et al. [23], and Deng et al [24] have already used or proposed zircon Hf-isotopic mapping as a method for metal exploration; yet the title still argues that the mapping is a "new method for exploration".  Perhaps the title should be changed again to "Zircon Hf-isotopic mapping applied to metal exploration of the Sanjiang Tethyan Orogenic Belt, Southwestern China."  The revised title would be much more accurate and would be shorter.  

Author Response

Point 1: why it is that high zircon Hf isotopic values correlate with juvenile crust and porphyry-copper deposits while low negative values correlate with ancient crust and magmatic hydrothermal tin-tungsten deposits.  Lines 90 to 92 are not enough.  A more in-depth explanation is needed.  Why is the Hf-isotopic content and the base-metal content of juvenile crust so different from ancient crust? 

 Response 1: Thank you for your comments .The manuscript has been thoroughly rewritten.Details are as follows:

Ore-bearing porphyries magmas forming the juvenile lower crust in the Sanjiang Tethyan Orogenic Belt were derived from asthenospheric mantle wedge metasomatized by upwelling of the asthenosphere in a post-subduction and post- collision settings[17]. Underplating of these magmas at the bottom of the crust inevitably led to the enrichment of juvenile lower crust in Cu(-Mo-Au) contents (Figure 5) [16]. (See Lines in 280-284)

Late Triassic abundant I-type granites and related porphyry Cu–Mo deposits ( e.g. Xuejiping (No. 10), Pulang (No. 11)) occur in the southern portion of the Yidun Arc (Figure 3 and Table 1).The porphyry Cu mineralization in the southern portion of the Yidun Arc region most likely originated from the enriched mantle wedge metasomatized by subduction-derived fluids and sediments[55,177]. The subduction derived fluids and sediments not only provided the volatile components (H2O and Cl), but also controlled the high oxygen fugacity (fO2 ), which makes calc-alkaline magmas favorable for porphyry Cu mineralization in this region[55,177]. (See Lines in 292-299)

Asthenosphere material upwelled, the temperature of the lower part of the rock group increased, and the thickened lower crust was partially melted to form [55]. During partial melting of the lower crust, chalcophile elements (such as Cu) were removed but siderophile metals (such as Mo) were left as residue in the cumulate zone of the deep arc crust and/or in the metasomatized mantle lithosphere. This mechanism facilitated porphyry Mo- Cu mineralization in the southern Yidun Terrane[158] (See Lines in 398-404)

The Sn-bearing granites are produced by regional tectonics driven by micro-plate subduction and collision [117]. At present, most of the discovered deposits in the Sn belt in western Yunnan are related to the remodeling-remelting granite magmatic activity [117]. The remodeling-remelting of the crust enriches the metallogenic metal elements initially dispersed in the crust (mantle). (See Lines in 417-422)

Point 2: Perhaps the title should be changed again to "Zircon Hf-isotopic mapping applied to metal exploration of the Sanjiang Tethyan Orogenic Belt, Southwestern China."  The revised title would be much more accurate and would be shorter. 

Response 2: Thank you for your comments. The title has been changed to “Zircon Hf-isotopic mapping applied to metal exploration of the Sanjiang Tethyan Orogenic Belt, Southwestern China.”

Reviewer 4 Report

The authors responded to all comments. Explanations are given about the impossibility of making some changes. Corrections are not always made to the desired extent. However, the article has been corrected and can be published. 

Author Response

Point : The authors responded to all comments. Explanations are given about the impossibility of making some changes. Corrections are not always made to the desired extent. However, the article has been corrected and can be published. 

Response : Thank you for your comments.
